# NorBench – A Benchmark for Norwegian Language Models

**David Samuel,**[1] **Andrey Kutuzov,**[1] **Samia Touileb,**[2] **Erik Velldal,**[1] **Lilja Øvrelid,**[1]
**Egil Rønningstad,**[1] **Elina Sigdel**[1] **and Anna Palatkina**[1]

[1]University of Oslo, Language Technology Group
[2]University of Bergen, MediaFutures

## Abstract

We present NorBench: a streamlined suite of NLP tasks and probes for evaluating Norwegian language models (LMs) on standardized data splits and evaluation metrics. We also introduce a range of new Norwegian language models (both encoder and encoder-decoder based). Finally, we compare and analyze their performance, along with other existing LMs, across the different benchmark tests of NorBench.

## 1 Introduction

This paper provides a suite of standardized tasks and probes for benchmarking of Norwegian language models (LMs). In addition to collecting a broad range of annotated datasets, we provide precise task definitions, pre-defined data splits and evaluation metrics, with corresponding scripts for streamlining the entire benchmarking pipeline. The resulting resource is dubbed NorBench. We furthermore present a range of new transformer-based language models (LMs) for Norwegian, trained with optimized configurations and architectures, and on different corpora with different pre-processing. Our contributions are as follows:

1. We introduce NorBench, a collection of Norwegian datasets and evaluation scripts that ensures simple, fair and standardized comparison between Norwegian LMs. The existing models from prior work are evaluated and compared. All data and code related to NorBench are publicly available online.[1]

2. An integral part of NorBench is diagnostic set of tasks that probe the affinity of LMs towards gender-bias and toxic language – an unfortunate side-effect for many models pre-trained on large amounts of text.

---

[1]https://github.com/ltgoslo/norbench

| Task | Train | Dev | Test |
|---|---|---|---|
| **Morpho-syntactic token-level tasks** | | | |
| Tokens in UD tasks | 489 217 | 67 619 | 54 739 |
| Named entities | 23 071 | 2 942 | 2 393 |
| **Sentiment analysis** | | | |
| SA documents | 34 903 | 4 360 | 4 351 |
| SA sentences | 7 973 | 1 411 | 1 181 |
| SA targets | 5 044 | 877 | 735 |
| **Linguistic acceptability** | | | |
| NoCoLA sentences | 116 195 | 14 289 | 14 383 |
| **Question answering** | | | |
| NorQuAD questions | 3 808 | 472 | 472 |
| **Machine translation** | | | |
| Bokmål–Nynorsk sentences | 10 000 | 10 000 | 10 000 |

Table 1: Number of labeled entities in the training, development, and test splits in the datasets used for the NorBench tasks.

3. We develop a new generation of Norwegian LMs – NorBERT$_3$ and NorT5 – that achieve state-of-the-art performance across most NorBench tasks. We provide multiple sizes of these models and show that even small versions maintain competitive performance.

4. We empirically test the impact of different available Norwegian training corpora on the downstream performance. Our results suggest that pre-training on a simple concatenation of all available resources is not always beneficial.

The rest of the paper is structured as follows. Section 2 provides an overview of the tasks included in NorBench. In Section 3, the details of our evaluation workflow are outlined. The architecture and the training collections of our novel LMs are described in Section 4, while in Section 5, we summarize and analyze the benchmarking results. Section 6 briefly describes prior work, while we point out directions for future work in Section 7, before concluding in Section 8.

## 2 NorBench task descriptions

We here briefly describe each task and associated dataset. The number of training examples for the different datasets and their train, development, and test splits are provided in Table 1. For the full details about each task, we refer the reader to the NorBench GitHub repository. Before we describe the various tasks, we first briefly comment on the two official varieties of written Norwegian.

**Bokmål and Nynorsk** Norwegian has two official written standards: *Bokmål* (BM), used by 85–90% of the Norwegian population, and *Nynorsk* (NN). While they are closely related, there can be relatively large lexical differences. The contextualised LMs presented in this paper are therefore trained jointly on both varieties, but with the minority variant Nynorsk represented by comparatively less data than Bokmål (reflecting the natural usage). Several previous studies have indicated that joint modeling of Bokmål and Nynorsk works well for many NLP tasks, like tagging and parsing (Velldal et al., 2017) and NER (Jørgensen et al., 2020). In cases where the labeled data for our benchmark tasks are available as separate versions for Bokmål and Nynorsk, we fine-tune models jointly on the combined BM/NN data. One practical advantage of training joint and 'variety agnostic' models, is that only a single model needs to be maintained, and we bypass the need for a separate 'language identification' step.

### 2.1 Morpho-syntactic token-level tasks

**UD tasks** We use the Norwegian Universal Dependencies Treebank (Øvrelid and Hohle, 2016; Velldal et al., 2017) from UD 2.11,[2] in turn converted from NDT (Solberg et al., 2014). In order to evaluate the general performance on Norwegian, we concatenate the Bokmål (BM) and Nynorsk (NN) datasets for both fine-tuning and evaluation. The models are challenged to predict universal part-of-speech tags (UPOS), universal features (UFeats), lemmas and dependency trees (Nivre et al., 2016).

*UPOS* tags cover the basic POS categories (17 tags) and *UFeats* differentiate more fine-grained lexical and grammatical properties of words, e.g. number, gender, and tense (172 tags in total). Both tagging tasks use the standard accuracy metric.

*Lemmatization* evaluates how well a language model understands Norwegian morphology and in order to transform an inflected word into its lemmatized form. An integral part of lemmatization – in our variety-agnostic setting – is implicit language identification, because Bokmål and Nynorsk have different lemmatization standards. Correct prediction requires exact match with the gold lemma; we report the aggregated token-wise accuracy.

*Dependency parsing* involves identifying the relationships between words in a sentence, resulting in a dependency tree that represents the grammatical structure of the sentence. By evaluating the quality of dependency parsing outputs by a language model, one can determine its ability to recognize and categorize the grammatical roles of words based on their syntactic function. We report the labeled attachment score (LAS), the standard evaluation metric for dependency parsing.[3]

**Named entity recognition** We use the NorNE[4] dataset which annotates the UD/NDT (for both Bokmål and Nynorsk) with named entities (Jørgensen et al., 2020). We predict 8 entity types: Person (PER), Organization (ORG), Location (LOC), Geo-political entity, with a locative sense (GPE-LOC), Geo-political entity, with an organization sense (GPE-ORG), Product (PROD), Event (EVT), and Nominals derived from names (DRV). The evaluation metric used is 'strict' micro $F_1$, requiring both the correct entity type and exact match of boundary surface string. It is computed using the code for the SemEval'13 Task 9.[5]

### 2.2 Sentiment analysis

**Document-level ternary polarity classification** The Norwegian Review Corpus (NoReC; 2nd release) (Velldal et al., 2018) comprises 43 425 professional reviews from a range of Norwegian news sources, and covering a range of different domains (e.g., books, movies, games, music, various consumer goods, etc.). The average length of a document is 389 words. While the reviews originally come with numerical ratings on a scale of 1–6, we here conflate these to three classes; *negative* (ratings 1–3), *fair* (4), and *positive* (5–6). This mapping is done to avoid problems with too few examples for the ratings in the extreme ends of the numerical scale. The dataset comes with prede-

---

[2] http://hdl.handle.net/11234/1-4923

[3] https://universaldependencies.org/conll18/evaluation.html
[4] https://github.com/ltgoslo/norne
[5] https://github.com/davidsbatista/NER-Evaluation

fined data splits (chronologically sorted), and we evaluate using macro $F_1$.

**Sentence-level ternary sentiment classification** We include the dataset NoReC*sentence*[6] for training and evaluating on the task of sentence-level polarity classification with respect to three classes (positive, negative, or neutral). As described by Kutuzov et al. (2021), this data is derived from NoReC*fine* (Øvrelid et al., 2020), a subset of NoReC, by aggregating the fine-grained annotations to the sentence-level, removing sentences with mixed sentiment. The evaluation metric is macro $F_1$.

**Targeted sentiment analysis** We use the NoReC*tsa*[7] dataset for the task of targeted sentiment analysis (TSA). As described in Rønningstad et al. (2022), the data is derived from NoReC*fine* by only including target expressions and the associated positive/negative polarity. The task is to jointly predict the target spans and their polarity, and we use the same evaluation strategy as for NER.

## 2.3 Linguistic acceptance

**NoCoLA** Norwegian corpus of linguistic acceptance (NoCoLA; Jentoft and Samuel, 2023) is used to evaluate language models on their understanding of Norwegian grammaticality. NoCoLA is derived from the ASK Corpus – a language learner corpus of Norwegian as a second language (Tenfjord et al., 2006), which contains texts written exclusively in Norwegian Bokmål, not covering the Nynorsk variety. We report the Matthews correlation coefficient (MCC; Matthews, 1975) on NoCoLA*class*, the official binary sentence classification variant of the dataset.

## 2.4 Question answering

**NorQuAD** is a Norwegian extractive question answering dataset which consists of 4 752 manually created question-answer pairs based on Wikipedia and news articles (Ivanova et al., 2023).[8] We here report token-level F1; human performance on the test portion of the dataset has been measured at 91.1% F1 (Ivanova et al., 2023).

## 2.5 Machine translation

**Bokmål–Nynorsk translation.** The fact that a monolingual Norwegian language model is actu-

ally trained on two language varieties – Bokmål and Nynorsk – allows us to evaluate generative models on machine translation. We collect the available Bokmål–Nynorsk bitexts,[9] deduplicate the sentences on both sides and split them into training, development, and test portions, each with 10 000 parallel sentences. We evaluate the translation from Bokmål to Nynorsk using SacreBLEU (Lin and Och, 2004; Post, 2018).[10]

## 2.6 Diagnostics of harmful predictions

Unlike the previous items, this is not a 'task', but rather a description of important model properties. We follow previous works on Norwegian to probe our language models for gender bias in occupations, as well as assessing the harmfulness of their sentence completions (Touileb et al., 2023, 2022; Touileb and Nozza, 2022).

# 3 NorBench baseline methodology

Below we describe various choices pertaining to fine-tuning the LMs for the various tasks. Note that, all of the approaches described here should be considered baselines, in the sense that the goal is not to produce state-of-the-art results, but rather to implement simple evaluation approaches allowing for a fair comparison of different LMs across the various tasks.

## 3.1 A joint model for UD tasks

Since the UD tasks are annotated within the same dataset, we evaluate them jointly with a single multi-task model. We follow the multi-task setup from UDify (Kondratyuk and Straka, 2019): first, we take a separate weighted convex combination of hidden layers for every subtask. Then, we average-pool these contextualized subword representations to get a vector embedding for each token. Finally, these vectors are input to classification heads for UPOS and UFeats tagging, to a classification head for predicting lemma transformation rules, and to biaffine attention heads for dependency parsing (Dozat and Manning, 2017).

---

[6]https://github.com/ltgoslo/norec_sentence
[7]https://github.com/ltgoslo/norec_tsa
[8]https://github.com/ltgoslo/NorQuAD

[9]Provided by the National Library of Norway: https://www.nb.no/sprakbanken/ressurskatalog/oai-nb-no-sbr-78/, https://www.nb.no/sprakbanken/ressurskatalog/oai-nb-no-sbr-47/

[10]The SacreBLEU metric involves several parameters that change the outcomes (Post, 2018), we use BLEU with no smoothing, 13a tokenization and no lowercasing – the default values in torchmetrics 0.11.4: https://torchmetrics.readthedocs.io/en/stable/text/sacre_bleu_score.

## 3.2 Text classification

For document- and sentence-level sentiment analysis, together with classification of linguistic acceptability, we utilize the same text classification approach, based on the widely-used fine-tuning scheme from Devlin et al. (2019). There, every tokenized text sequence is prefixed by a special `[CLS]` token, appended by a `[SEP]` token and passed into a pre-trained language model, which produces a contextualized representation for the special `[CLS]` token. Finally, this representation is passed into the downstream classifier that produces the final prediction among the available classes. For the encoder-decoder models, we chose three target words as the class labels ('negativ', 'nøytral' and 'positivt') and fine-tuned the models to generate these target words given the input text. At the inference time, an input text is assigned a class depending on whether the corresponding target word occurs in the generated text.

## 3.3 Sequence labeling for NER and TSA

NER and TSA are approached as a sequence labeling task where we classify text spans by tagging tokens with beginning-inside-outside tags (BIO; Ramshaw and Marcus, 1995).

## 3.4 Question answering

We follow the SQuAD fine-tuning method introduced in BERT (Devlin et al., 2019). For every question and context passage, the goal is to identify the answer within the passage. The question and passage texts are concatenated together and the evaluated model is trained to predict the first and last token of the answer – the problem is cast as 2-task binary classification problem.

## 3.5 Machine translation

We use this task only for evaluation of generative sequence-to-sequence models such as T5s (Raffel et al., 2020). This task naturally fits these models – the source sentence is encoded and the target sentence is decoded with the respective parts of the model. We use simple greedy decoding for generation during inference.

## 3.6 Probing for gender-bias and harmfulness

We take advantage of the MLM objective of the models, and create templates consisting of gendered head-words, followed by predicates.

**Gender-bias**   To probe for gender bias in occupations, we follow and use the templates of Touileb et al. (2023) and Touileb et al. (2022). These templates are sequences of masked gendered head-words (*e.g.* the woman, the man, the sisters, the uncles ...), followed by predicates pertaining to verbs related to performing an occupation (*e.g.* is, was, worked as, ...), followed by a set of occupations extracted from the Norwegian Statistics bureau (Touileb et al., 2022). Using the probabilities of the masked gendered head-words, we compute the two bias scores: normative and descriptive as defined by Touileb et al. (2023). The normative score compares the gender-based aggregated probability distributions of templates with a normative distribution of genders in occupations. The idea here is that genders should be equally represented in each occupation with a gender distribution falling between 45% and 55% for each occupation. The descriptive bias score compares the probability distribution of genders across occupations as represented in language models, to the real world distribution of these genders based on the Norwegian Statistics bureau data.

**Harmfulness**   We also follow Touileb and Nozza (2022) to assess the harmfulness of sentence-completions of each language model. We use their templates, constructed similarly to the previous templates where the head-words are gendered-nouns followed by predicates as defined by Nozza et al. (2021), and where the last token is masked. The probing is therefore aiming at completing sentences, by looking at top one, five, ten, and twenty most likely words for each template. Once the completions returned by the models, we compute the HONEST score (Nozza et al., 2021). This score is a word-level completion score that maps generated completions to their respective language-specific HurtLex (Bassignana et al., 2018) lexicon of offensive words. The scores represent the total number of completions existing in the lexicon compared to the total amount of returned completions.

## 4   New Norwegian language models

A number of large Norwegian language models have appeared in recent years: to name only the masked LMs, Kutuzov et al. (2021) trained NorBERT$_1$, followed by NorBERT$_2$, and Kummervold et al. (2021) introduced NB-BERT models, coming in different sizes. In this paper, we present a set of novel masked and text-to-text LMs for

Norwegian trained according to the LTG-BERT training recipe by Samuel et al. (2023). We dub these models **NorBERT₃** and **NorT5** and evaluate their performance across different model sizes and training corpora.

## 4.1 Training corpora

**Text sources**   Our LM training dataset included the following text collections:

- Norwegian Wikipedia dumps (BM/NN) from October 2022; about 180 million words;
- NBDigital, public domain texts released by the National Library (NB) of Norway in 2015; 660 million words;[11]
- Norwegian News Corpus (NAK): a collection of Norwegian news texts (both Bokmål and Nynorsk) published between 1998 and 2019; 1.7 billion words;[12]
- Norwegian Colossal Corpus (NCC): the public part of the large and heterogenous corpus released by NB in 2022[13] (Kummervold et al., 2021); about 6.9 billion words;
- Norwegian part of web-crawled mC4 corpus (Xue et al., 2021); about 15 billion words.

The 'standard' models were trained on the concatenation of these corpora, yielding a training collection of about 25 billion word tokens. In Section 5.2, we investigate the effects of 'oversampling' higher-quality sources and training separate models from scratch on NAK, NCC, mC4, Wikipedia, and NBDigital.

**Deduplication**   Before training, all the corpora were de-duplicated on the paragraph-level, using SimHash[14] and removing exact duplicates. The same was done across corpora, reducing their size up to 10%, depending on the corpus.

**Filtering**   Since the largest portion of our training corpus is sourced from web-crawled texts, it is crucial to filter out any unnatural language. Even though our main web-text source is the multilingual Colossal *Clean* Crawled Corpus (mC4), it still contains noisy texts (Dodge et al., 2021), which was

---

[11] https://www.nb.no/sprakbanken/en/resource-catalogue/oai-nb-no-sbr-34/
[12] https://www.nb.no/sprakbanken/ressurskatalog/oai-nb-no-sbr-4/
[13] https://huggingface.co/datasets/NbAiLab/NCC
[14] https://github.com/ChenghaoMou/text-dedup

---

| Hyperparameter | x-small | small | base | large |
|---|---|---|---|---|
| Number of parameters | 15M | 40M | 123M | 353M |
| Number of layers | 12 | 12 | 12 | 24 |
| Hidden dimension | 192 | 384 | 768 | 1 024 |
| Attention heads | 3 | 6 | 12 | 16 |

Table 2: The main hyperparameters of our four configurations of NorBERT₃ language models. Full list of hyperparameters is given in Table 9.

also apparent when we manually investigated some of the Norwegian samples. We follow the filtering heuristics implemented for the MassiveText corpus (Rae et al., 2021) and adapt them for Norwegian.

## 4.2 Architecture

We employ the masked language modeling approach for pre-training NorBERT₃ language models and follow the optimized training method from Samuel et al. (2023). This approach differs from the standard BERT (Devlin et al., 2019) training as follows:

1. Liu et al. (2019) found out that BERT is undertrained and the next-sentence prediction task is unnecessary – we thus pre-train for $8\times$ more steps, use sequence length of 512 throughout the whole training, and utilize only the masked language modeling task (MLM) without next-sentence prediction.

2. SpanBERT (Joshi et al., 2020) and T5 (Raffel et al., 2020) demonstrated the advantages of masking random spans instead of individual subwords as in Devlin et al. (2019). Thus, for our MLM objective, the data loader iteratively samples random spans until 15% of the input text is masked. The length of each span is sampled from $\text{Geo}(p)$, where $p = 1/3$.

3. Samuel et al. (2023) compared various configurations of transformer architectures and of the training hyperparameters. We employ the best performing setting for our pre-training. Crucial upgrades involve using the NormFormer layer normalization (Shleifer and Ott, 2022), disentangled attention with relative positions (He et al., 2021) and increased amount of weight decay. Please refer to Samuel et al. (2023) for more pre-training details.

**Parameter count**   We train four LMs of different sizes (Table 2), accommodating users with vary-

ing degrees of computational resources, and to establish a baseline performance across LMs with different number of parameters.

**Vocabulary and tokenizer** We utilize Word-Piece subword tokenizer (Wu et al., 2016) and set its vocabulary size to 50 000. Following GPT-2 (Radford et al., 2019), we represent the text as a sequence of UTF-8 bytes instead of Unicode characters, which substantially reduces the number of out-of-vocabulary tokens. We train the tokenizer on the full corpus utilizing the open implementation from the `tokenizers` library.[15]

**NorT5** Some NLP tasks, for example machine translation, require a generative language model. Thus we extend the encoder-only architecture of NorBERT$_3$ to full encoder-decoder transformer and pre-train the resulting model, dubbed NorT5, on text-to-text masked language modeling (T5; Raffel et al., 2020). We use the same text corpus, tokenizer and training settings as in NorBERT$_3$ when applicable. For the T5-specific training choices, we follow T5 version 1.1 – i.e. pre-training only on self-supervised masked LM and no parameter sharing between the embedding and classifier layer.[16]

### 4.3 Pre-training details

In order to reduce training time, pre-training is parallelized over multiple GPUs with the global batch size of 8 192. The number of GPUs used depends on the size of pre-trained language models, ranging between 16 and 512 AMD Instinct MI250X GPUs, each with 128GB memory. The amount of training steps is 250 000, increasing the training budget of the original BERT models 8 times. NorBERT$_{3,\text{ base}}$ was pre-trained in 280 hours using this setting.

## 5 Benchmarking results

In addition to our NorBERT$_3$ models, we also benchmark these existing models:

- *BERT* (Devlin et al., 2019): to get a baseline performance, we include the scores of an *English*-only language model. Its scores suggest how much information can be inferred from the supervised datasets without any understanding of Norwegian.

- *mBERT* (Devlin et al., 2019): multilingual BERT pre-trained on 104 languages, including Norwegian. The training was done exclusively on Wikipedia dumps with oversampled texts from lower resource languages.

- *XLM-R* (Conneau et al., 2020): more advanced multilingual LM that outperformed mBERT on most tasks. XLM-R models were trained on CommonCrawl data for 100 languages.

- *NB-BERT* (Kummervold et al., 2021): NB-BERT$_{\text{base}}$ model utilized a warm start from pre-trained mBERT. It was later followed by NB-BERT$_{\text{large}}$ trained from scratch on Norwegian data. Both models are trained on the full – i.e., partially non-public – NCC corpus.

- *NorBERT$_1$ and NorBERT$_2$* (Kutuzov et al., 2021): both models follow the pre-training approach of the original BERT model (Devlin et al., 2019). NorBERT$_1$ is pre-trained on NAK and dumps from both Norwegian Wikipedias, and NorBERT$_2$ utilizes the Norwegian part of mC4 and the public part of NCC.

- *ScandiBERT*: Scandinavian BERT trained on a combination of Danish, Faroese, Icelandic, Norwegian, and Swedish texts. However, more than 60% of the training corpus consists of texts from the Norwegian NCC.[17]

Our NorT5 models are compared with the multilingual T5 models (mT5; Xue et al., 2021) and with a set of so-called North-T5 models – mT5 models further fine-tuned solely on Norwegian (published online in 2022).[18]

### 5.1 Comparison of models

Table 3 and Table 4 show results across all the current NorBench tasks for all language models described above (sorted by their size in the number of parameters). Note that we deliberately do not report any average score across all tasks, since we believe that such aggregated scores do not contribute to real understanding of strong and weak sides of different models: one should pay attention to the performance in particular tasks of interest.

**Encoder-only scores** Not surprisingly, one can see that it is the largest monolingual models that tend to perform best across the board, and

[15]https://github.com/huggingface/tokenizers
[16]https://github.com/google-research/text-to-text-transfer-transformer/blob/main/released_checkpoints.md#t511

[17]The training procedure is briefly described here; https://huggingface.co/vesteinn/ScandiBERT.
[18]https://huggingface.co/north/t5_base_NCC

| Model | Size | UPOS | UFeats | Lemma | LAS | NER | Doc. SA | Sent. SA | TSA | NoCoLA | NorQuAD |
|---|---|---|---|---|---|---|---|---|---|---|---|
| NorBERT$_{3,\text{ x-small}}$ | 15M | **98.8**$^{\pm0.1}$ | **97.0**$^{\pm0.1}$ | **97.6**$^{\pm0.1}$ | **92.2**$^{\pm0.1}$ | **86.3**$^{\pm0.4}$ | 69.6$^{\pm2.4}$ | **66.2**$^{\pm1.2}$ | 43.2$^{\pm0.5}$ | **47.1**$^{\pm0.5}$ | **65.6**$^{\pm3.9}$ |
| NorBERT$_{3,\text{ small}}$ | 40M | **98.9**$^{\pm0.0}$ | **97.9**$^{\pm0.0}$ | **98.3**$^{\pm0.1}$ | **93.7**$^{\pm0.0}$ | **89.0**$^{\pm0.3}$ | **74.4**$^{\pm0.5}$ | **71.9**$^{\pm1.3}$ | **48.9**$^{\pm0.9}$ | **55.9**$^{\pm0.2}$ | **80.5**$^{\pm1.2}$ |
| BERT$_{\text{base, cased}}$ | 111M | 97.9$^{\pm0.0}$ | 96.4$^{\pm0.1}$ | 97.9$^{\pm0.0}$ | 89.8$^{\pm0.2}$ | 73.4$^{\pm0.7}$ | 57.3$^{\pm1.4}$ | 53.0$^{\pm1.1}$ | 23.2$^{\pm2.2}$ | 23.9$^{\pm0.4}$ | 44.9$^{\pm2.2}$ |
| NorBERT$_1$ | 111M | 98.8$^{\pm0.0}$ | 97.8$^{\pm0.0}$ | 98.5$^{\pm0.0}$ | 93.3$^{\pm0.1}$ | 86.9$^{\pm0.9}$ | 70.1$^{\pm0.4}$ | 70.7$^{\pm0.9}$ | 45.4$^{\pm1.1}$ | 35.9$^{\pm1.7}$ | 72.5$^{\pm1.6}$ |
| NorBERT$_{3,\text{ base}}$ | 123M | **99.0**$^{\pm0.0}$ | **98.3**$^{\pm0.1}$ | 98.8$^{\pm0.0}$ | **94.2**$^{\pm0.1}$ | 89.4$^{\pm0.9}$ | **76.2**$^{\pm0.8}$ | **74.4**$^{\pm0.3}$ | **50.2**$^{\pm0.7}$ | **59.2**$^{\pm0.3}$ | **86.2**$^{\pm0.3}$ |
| NorBERT$_2$ | 125M | 98.7$^{\pm0.0}$ | 97.6$^{\pm0.0}$ | 98.2$^{\pm0.0}$ | 93.4$^{\pm0.1}$ | 85.0$^{\pm0.9}$ | 73.5$^{\pm1.1}$ | 72.5$^{\pm1.5}$ | 45.4$^{\pm1.1}$ | 56.1$^{\pm0.3}$ | 76.6$^{\pm0.7}$ |
| ScandiBERT | 124M | 98.9$^{\pm0.0}$ | 98.1$^{\pm0.0}$ | 98.7$^{\pm0.0}$ | **94.1**$^{\pm0.1}$ | **89.4**$^{\pm0.5}$ | 73.9$^{\pm0.4}$ | 71.6$^{\pm1.3}$ | 48.8$^{\pm1.0}$ | 57.1$^{\pm0.4}$ | 79.0$^{\pm0.7}$ |
| NB-BERT$_{\text{base}}$ | 178M | 98.9$^{\pm0.0}$ | **98.3**$^{\pm0.0}$ | **98.9**$^{\pm0.0}$ | **94.1**$^{\pm0.1}$ | **89.6**$^{\pm0.9}$ | 74.3$^{\pm0.6}$ | 73.7$^{\pm0.8}$ | 49.2$^{\pm1.3}$ | 58.1$^{\pm0.5}$ | 79.1$^{\pm1.2}$ |
| mBERT | 178M | 98.4$^{\pm0.0}$ | 97.3$^{\pm0.1}$ | 98.3$^{\pm0.0}$ | 92.2$^{\pm0.1}$ | 83.5$^{\pm0.6}$ | 67.9$^{\pm1.2}$ | 62.7$^{\pm1.2}$ | 39.6$^{\pm1.3}$ | 46.4$^{\pm0.7}$ | 76.5$^{\pm0.9}$ |
| XLM-R$_{\text{base}}$ | 278M | 98.8$^{\pm0.0}$ | 97.7$^{\pm0.0}$ | 98.7$^{\pm0.0}$ | 93.7$^{\pm0.1}$ | 87.6$^{\pm0.6}$ | 73.1$^{\pm0.7}$ | 72.2$^{\pm0.3}$ | 49.4$^{\pm0.5}$ | 58.6$^{\pm0.3}$ | 78.9$^{\pm0.6}$ |
| NorBERT$_{3,\text{ large}}$ | 353M | **99.1**$^{\pm0.0}$ | **98.5**$^{\pm0.0}$ | **99.1**$^{\pm0.0}$ | **94.6**$^{\pm0.1}$ | **91.4**$^{\pm0.5}$ | **79.2**$^{\pm0.7}$ | **78.4**$^{\pm0.6}$ | 54.1$^{\pm0.6}$ | **61.0**$^{\pm0.4}$ | **88.7**$^{\pm0.8}$ |
| NB-BERT$_{\text{large}}$ | 355M | 98.7$^{\pm0.0}$ | 98.2$^{\pm0.1}$ | 98.3$^{\pm0.1}$ | **94.6**$^{\pm0.1}$ | 89.8$^{\pm0.6}$ | **79.2**$^{\pm0.9}$ | 77.5$^{\pm0.7}$ | **54.6**$^{\pm0.7}$ | 59.7$^{\pm0.1}$ | 87.0$^{\pm0.5}$ |
| XLM-R$_{\text{large}}$ | 560M | 98.9$^{\pm0.0}$ | 98.0$^{\pm0.0}$ | 98.8$^{\pm0.1}$ | 94.3$^{\pm0.1}$ | 87.5$^{\pm1.0}$ | 76.8$^{\pm0.6}$ | 75.4$^{\pm1.3}$ | 52.3$^{\pm0.6}$ | 58.6$^{\pm0.3}$ | 84.8$^{\pm0.5}$ |

Table 3: NorBench scores for the existing language models and our novel NorBERT$_3$ family of models. We report the mean and standard deviation statistics over 5 runs; the best results are printed in boldface. The 'Size' column reports the number of parameters in the model; the models are sorted by this value and divided into four size categories. The best results (within one standard deviation) in each category are typeset in bold.

| Model | Size | Doc. SA | Sent. SA | NoCoLA | NB-NN |
|---|---|---|---|---|---|
| NorT5$_{\text{x-small}}$ | 32M | **70.1**$^{\pm1.1}$ | **55.2**$^{\pm13.6}$ | **51.4**$^{\pm0.4}$ | **82.1**$^{\pm0.2}$ |
| NorT5$_{\text{small}}$ | 88M | **73.7**$^{\pm1.4}$ | **73.2**$^{\pm0.7}$ | **54.4**$^{\pm0.3}$ | **85.1**$^{\pm0.1}$ |
| mT5$_{\text{small}}$ | 300M | 24.8$^{\pm3.0}$ | 22.4$^{\pm0.0}$ | 25.4$^{\pm5.4}$ | 33.2$^{\pm0.3}$ |
| North-T5$_{\text{small}}$ | 300M | 20.9$^{\pm0.1}$ | 22.4$^{\pm0.0}$ | 33.8$^{\pm7.9}$ | 36.0$^{\pm0.1}$ |
| T5$_{\text{base}}$ | 223M | 47.2$^{\pm3.5}$ | 41.3$^{\pm3.2}$ | 17.6$^{\pm0.8}$ | 8.9$^{\pm0.0}$ |
| NorT5$_{\text{base}}$ | 228M | **77.4**$^{\pm0.4}$ | **73.4**$^{\pm0.8}$ | **58.9**$^{\pm0.3}$ | **86.6**$^{\pm0.1}$ |
| mT5$_{\text{base}}$ | 582M | 21.0$^{\pm0.1}$ | 24.8$^{\pm4.9}$ | 25.3$^{\pm10.1}$ | 38.6$^{\pm0.1}$ |
| North-T5$_{\text{base}}$ | 582M | 21.2$^{\pm0.3}$ | 22.5$^{\pm0.2}$ | 41.1$^{\pm9.6}$ | 39.8$^{\pm0.2}$ |
| NorT5$_{\text{large}}$ | 808M | **77.7**$^{\pm0.5}$ | **76.9**$^{\pm2.0}$ | **59.4**$^{\pm0.5}$ | **86.8**$^{\pm0.1}$ |
| mT5$_{\text{large}}$ | 1 230M | 59.9$^{\pm20.1}$ | 29.1$^{\pm6.6}$ | 50.4$^{\pm4.0}$ | 40.0$^{\pm0.1}$ |
| North-T5$_{\text{large}}$ | 1 230M | 72.9$^{\pm1.2}$ | 22.4$^{\pm0.0}$ | 46.8$^{\pm18.7}$ | 41.1$^{\pm0.1}$ |

Table 4: NorBench scores for encoder-decoder models, evaluated in a generative text-to-text setting. The best results (within one standard deviation) in each category are typeset in bold.

NorBERT$_{3,\text{ large}}$ (with 353M parameters) specifically obtains the highest scores for most of the tasks except targeted sentiment analysis. At the same time, we see that the smaller models are still very competitive – perhaps most notably NorBERT$_{3,\text{ small}}$ (with 40M parameters) – and there is certainly an aspect of diminishing returns with increasing the number of parameters.

**Encoder-decoder scores** Table 4 shows the results of T5 models evaluated on four generative tasks. We can see that the performance monotonously improves with scale but the differences between models of different sizes are not drastic. Unfortunately, we found the mT5-based models to be highly unstable and unable to reach decent performance. Our NorT5-large model turned out to be the best across all the tasks.

**Gender-bias evaluation** Table 5 shows the normative and descriptive occupational bias scores for each model. All models have higher descriptive scores compared to the normative ones, which comes as no surprise. Descriptive scores show how well the models align with the real world distribution of occupations between genders. While no model achieves a perfect score, the top three best models are the NorBERT$_{3,\text{ base}}$ trained on respectively Wikipedia, NAK, and NCC. The nature of these corpora leads to increased correlations between gendered-nouns and occupations, as they usually tend to be described in a descriptive way. NorT5$_{\text{x-small}}$ achieves the worst descriptive bias score of all models, but still scoring better than the best model on the normative score. Looking more specifically at gender-dominated and gender-neutral occupations, it is clear that all models are much better at identifying female-dominated occupations. All models achieve very low scores on gender-neutral occupations, suggesting a tendency to correlate occupations with one gender, rather then equally representing them. These results can be seen in Table 8 in the Appendix.

On the other hand, when we expect genders to be equally represented, no model achieves as high scores in the normative scores, as in the descriptive ones. The best Norwegian model (second best overall) is the smallest model NorBERT$_{3,\text{ x-small}}$,

| Model | Normative | Descriptive |
|---|---|---|
| NorBERT$_{3,\text{x-small}}$ | 19.78 | 37.36 |
| NorBERT$_{3,\text{small}}$ | 8.54 | 34.92 |
| NorBERT$_1$ | 16.23 | 39.31 |
| NorBERT$_2$ | 3.17 | 34.67 |
| NB-BERT$_{\text{base}}$ | 18.55 | 36.50 |
| ScandiBERT | 14.04 | 43.95 |
| mBERT | **24.66** | 41.88 |
| XLM-R$_{\text{base}}$ | 16.60 | 36.99 |
| NorBERT$_{3,\text{base}}$ | 13.55 | 39.43 |
| XLM-R$_{\text{large}}$ | 19.16 | 46.64 |
| NB-BERT$_{\text{large}}$ | 11.35 | 40.90 |
| NorBERT$_{3,\text{large}}$ | 13.67 | 42.73 |
| NorBERT$_{3,\text{base}}$, oversampled | 9.64 | 36.99 |
| NorBERT$_{3,\text{base}}$, NAK only | 14.04 | 49.81 |
| NorBERT$_{3,\text{base}}$, NCC only | 12.57 | 48.84 |
| NorBERT$_{3,\text{base}}$, mC4 only | 11.72 | 39.31 |
| NorBERT$_{3,\text{base}}$, NB only | 12.33 | 38.21 |
| NorBERT$_{3,\text{base}}$, Wiki only | 15.99 | **50.42** |
| NorT5$_{\text{x-small}}$ | 8.91 | 33.69 |
| NorT5$_{\text{small}}$ | 0.12 | 34.06 |
| NorT5$_{\text{base}}$ | 5.25 | 43.83 |
| NorT5$_{\text{large}}$ | 2.56 | 34.18 |

Table 5: Normative and descriptive occupational bias scores (Touileb et al., 2023). Best scores are typeset in bold, and worst scores are underlined.

| Model | k = 1 | k = 5 | k = 10 | k = 20 |
|---|---|---|---|---|
| NorBERT$_{3,\text{x-small}}$ | 0.0062 | 0.0062 | 0.0040 | 0.0037 |
| NorBERT$_{3,\text{small}}$ | 0.0015 | 0.0018 | 0.0027 | 0.0049 |
| NorBERT$_1$ | 0.0310 | 0.0378 | 0.0306 | 0.0258 |
| NorBERT$_2$ | 0.0356 | 0.0229 | 0.0189 | 0.0159 |
| NB-BERT$_{\text{base}}$ | 0.0124 | 0.0083 | 0.0080 | 0.0069 |
| ScandiBERT | **0.0** | 0.0010 | 0.0043 | 0.0045 |
| mBERT | **0.0** | 0.0028 | 0.0057 | 0.0068 |
| XLM-R$_{\text{base}}$ | 0.0450 | 0.0169 | 0.0117 | 0.0128 |
| NorBERT$_{3,\text{base}}$ | **0.0** | 0.0027 | 0.0026 | 0.0055 |
| XLM-R$_{\text{large}}$ | 0.0342 | 0.0158 | 0.0131 | 0.0116 |
| NB-BERT$_{\text{large}}$ | 0.0294 | 0.0285 | 0.0279 | 0.0244 |
| NorBERT$_{3,\text{large}}$ | **0.0** | 0.0006 | 0.0013 | 0.0033 |
| NorBERT$_{3,\text{base}}$, oversampled | 0.0046 | 0.0071 | 0.0085 | 0.0092 |
| NorBERT$_{3,\text{base}}$, NAK only | 0.0093 | 0.0080 | 0.0093 | 0.0125 |
| NorBERT$_{3,\text{base}}$, NCC only | **0.0** | 0.0006 | 0.0010 | 0.0028 |
| NorBERT$_{3,\text{base}}$, mC4 only | **0.0** | 0.0003 | **0.0009** | 0.0038 |
| NorBERT$_{3,\text{base}}$, NB only | 0.0015 | 0.0031 | 0.0012 | **0.0026** |
| NorBERT$_{3,\text{base}}$, Wiki only | **0.0** | 0.0012 | 0.0071 | 0.0082 |
| NorT5$_{\text{x-small}}$ | **0.0** | 0.0010 | 0.0018 | **0.0026** |
| NorT5$_{\text{small}}$ | **0.0** | 0.0003 | 0.0018 | 0.0037 |
| NorT5$_{\text{base}}$ | **0.0** | 0.0010 | 0.0077 | 0.0090 |
| NorT5$_{\text{large}}$ | **0.0** | **0.0** | 0.0014 | 0.0037 |

Table 6: The harmfulness score of models looking at top one, five, ten, and twenty most likely completions using HONEST (Nozza et al., 2021). The best scores are in bold, while the worst are underlined.

which might suggest that from a normative perspective, the smaller the model, the more balanced representation of genders, at least when it comes to occupations. The best scoring model is the multilingual mBERT model. On a closer analysis, it is apparent that mBERT is very good at correlating occupations with the male gender (similarly to the descriptive score in Table 8 in the Appendix), which seems to skew the metric. This might exhibit a weakness in the metric, where models skewed towards one gender can get higher overall scores even if they fail to represent the other gender.

**Harmfulness scores** In addition to the normative and descriptive occupational bias scores, we also compute the harmfulness of the sentence-completions generated by these models. Table 6 shows the HONEST scores (Nozza et al., 2021) of each model. Here we evaluate the top-k completions, where we look at the first, five, ten, and twenty most likely completions. Overall, NorBERT$_3$ and NorT5 models achieve very low harmfulness scores compared to the other Norwegian language models. All NorT5 models do not return harmful words as the most likely completions, and are overall generating few problematic

completions. However, since the HONEST score relies on lexicons, some completions not included in these might still be harmful. XLM-R$_{\text{base}}$ is the worst model in top one completions, while the NorBERT$_1$ is the worst model across all remaining top k completions.

## 5.2 Comparison of Norwegian corpora

The downstream performance of a language model is a result of a combination of training choices and choices of the training corpus. In order to study the second aspect, we fix the training configuration and pre-train multiple NorBERT$_{3,\text{base}}$ models on different Norwegian corpora.

We compare a simple concatenation of all available resources (*'combined'*) against a variant with oversampling the quality data. The reasoning behind this was that the mC4 corpus is the most noisy of all the above, since it is created by web crawling. We hypothesized that artificially increasing the amount of data from the cleaner corpora (Wikipedia, NBDigital, NCC and NAK) will improve the resulting model's performance. We implemented this by creating an *'oversampled'* train collection where all the sentences from the clean corpora were repeated twice, so that the total size of the 'clean' part approximately matched the size of

| Corpus | UPOS | UFeats | Lemma | LAS | NER | Doc. SA | Sent. SA | TSA | NoCoLA | NorQuAD |
|---|---|---|---|---|---|---|---|---|---|---|
| Combined | **99.0**$^{\pm0.0}$ | **98.3**$^{\pm0.1}$ | **98.8**$^{\pm0.0}$ | **94.2**$^{\pm0.1}$ | 89.4$^{\pm0.7}$ | 76.2$^{\pm0.8}$ | 74.4$^{\pm0.3}$ | **52.2**$^{\pm0.7}$ | **59.2**$^{\pm0.3}$ | **86.2**$^{\pm0.3}$ |
| Oversampled | 98.9$^{\pm0.0}$ | **98.2**$^{\pm0.0}$ | 98.7$^{\pm0.0}$ | 94.1$^{\pm0.1}$ | **90.5**$^{\pm0.3}$ | 75.0$^{\pm0.4}$ | 75.2$^{\pm0.5}$ | 50.4$^{\pm0.4}$ | 57.6$^{\pm0.1}$ | 83.4$^{\pm0.7}$ |
| NAK | 98.9$^{\pm0.0}$ | 98.0$^{\pm0.0}$ | 98.5$^{\pm0.0}$ | 94.1$^{\pm0.1}$ | 90.4$^{\pm0.6}$ | **76.9**$^{\pm0.1}$ | **77.5**$^{\pm0.9}$ | 51.3$^{\pm0.7}$ | 58.3$^{\pm0.3}$ | 82.5$^{\pm0.4}$ |
| NCC | **99.0**$^{\pm0.0}$ | **98.2**$^{\pm0.0}$ | 98.7$^{\pm0.0}$ | **94.3**$^{\pm0.1}$ | 89.5$^{\pm0.6}$ | 74.8$^{\pm0.3}$ | 74.8$^{\pm1.4}$ | 50.0$^{\pm0.5}$ | 58.3$^{\pm0.4}$ | 83.0$^{\pm1.2}$ |
| mC4 | **99.0**$^{\pm0.0}$ | 98.1$^{\pm0.0}$ | 98.7$^{\pm0.0}$ | **94.2**$^{\pm0.1}$ | 90.2$^{\pm0.5}$ | 76.3$^{\pm0.6}$ | 76.8$^{\pm0.7}$ | 50.8$^{\pm0.9}$ | 58.5$^{\pm0.3}$ | 83.2$^{\pm0.5}$ |
| Wiki | 98.9$^{\pm0.0}$ | 97.6$^{\pm0.0}$ | 98.3$^{\pm0.0}$ | 93.6$^{\pm0.1}$ | 87.9$^{\pm0.3}$ | 71.9$^{\pm1.0}$ | 68.9$^{\pm1.2}$ | 44.9$^{\pm0.4}$ | 54.1$^{\pm0.3}$ | 78.2$^{\pm0.5}$ |
| NBDigital | 98.9$^{\pm0.0}$ | 98.0$^{\pm0.0}$ | 98.7$^{\pm0.0}$ | 93.9$^{\pm0.1}$ | 87.1$^{\pm0.7}$ | 72.7$^{\pm0.4}$ | 70.1$^{\pm0.5}$ | 45.2$^{\pm0.9}$ | 56.1$^{\pm0.1}$ | 79.3$^{\pm0.6}$ |

Table 7: The downstream performance of NorBERT$_{3,\,base}$ models pre-trained on different corpora. We report the mean and standard deviation statistics over 5 runs; the best results (within one standard deviation) are shown in boldface.

the mC4 corpus. In addition, to study the respective usefulness of particular Norwegian text collections, we trained separate models from scratch on NAK, NCC, mC4, Wikipedia, and NBDigital.

**Corpora comparison results**  Table 7 shows the results. We believe there are two noteworthy – and perhaps surprising – take-aways:

1. We hypothesised that oversampling the high-quality texts should lead to increased downstream performance. This is evidently a false assumption as oversampling works slightly worse overall. Large language models are known to be sensitive to duplicate data (Lee et al., 2022), which might explain such a behavior.

2. A straightforward concatenation of all available resources does not necessarily lead to better performance – but it is a reasonable approach for a general model as it works the best on average. On the other hand, pre-training only on NAK leads to substantially improved performance on sentiment analysis, perhaps due to a closer match in terms of text type.

## 6   Related work

Evaluating pre-trained language models for particular languages and cross-linguely is a venerable research sub-field within NLP. Well-known benchmark sets for English include GLUE (Wang et al., 2018), SuperGLUE (Wang et al., 2019), and GLGE (Liu et al., 2021), among others. However, up to now benchmarking LMs for Norwegian was limited to separate test sets with non-standardised evaluation workflows. ScandEval (Nielsen, 2023) aims to create a standard natural language understanding benchmark across Scandinavian languages (Danish, Swedish, and Norwegian). However, it does not focus on evaluating specifically Norwegian tasks

and half of its Norwegian benchmarks (linguistic acceptability and question answering) are not human-annotated. We address this issue with Nor-Bench.

## 7   Future work

We consider NorBench to be a dynamic resource that we plan to continually extend in future work, to support additional tasks and additional architectures. While we anticipate including new annotated benchmark data as they may become available in the future, there are also existing datasets that we plan to include in the shorter term, like coreference resolution based on the NARC dataset (Mæhlum et al., 2022) and negation resolution based on NoReC$_{neg}$ (Mæhlum et al., 2021). Finally, we also plan on adding tasks that more specifically target generative models, including sequence-generation tasks like summarization, but also prompt-based formulations of the existing NorBench tasks for few-shot evaluation.

## 8   Summary

In this paper we have presented NorBench, a set of standardized benchmark tasks for systematically evaluating and comparing Norwegian language models. The aim of this effort is to provide NLP practitioners with a comprehensive and streamlined service, including a leaderboard, human-annotated datasets, evaluation workflow, and open code implementing this workflow.

This paper also describes and evaluates a set of novel NorBERT$_3$ masked LMs trained on several different Norwegian text collections in different model sizes. They are shown to outperform Norwegian LMs from prior work on the majority of NorBench tasks.

## Acknowledgements

NorBench forms part of NorLM, an initiative coordinated by the Language Technology Group (LTG) at the University of Oslo (UiO), with the goal of developing resources for large-scale language modeling for Norwegian. The efforts described in the current paper were jointly funded by the SANT project (Sentiment Analysis for Norwegian; coordinated by LTG at UiO and funded by the the Research Council of Norway, grant number 270908), and the HPLT project (High Performance Language Technologies; coordinated by Charles University). The computations were performed on resources provided through Sigma2 – the national research infrastructure provider for High-Performance Computing and large-scale data storage in Norway.

Parts of this work was supported by industry partners and the Research Council of Norway with funding to MediaFutures: Research Centre for Responsible Media Technology and Innovation, through the Centres for Research-based Innovation scheme, project number 309339.

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

## A Sentiment analysis classification – details

As mentioned above, initially reviews were rated on a scale from 1 to 6 but later we have narrowed down the classes to 3 (classes 1, 2, and 3 mapped to the 'negative' class, class 4 to the 'fair' class, and classes 5 and 6 to the 'positive' class).

Document-level sentiment analysis is complicated by 2 factors. First, 40% of all the dataset texts were longer than 512 (white-space separated) tokens with the maximum text length reaching 3 943 tokens. Second, the average text length as well as the number of samples in NoReC increased from negative to positive classes. Therefore we had a challenging task, with the 'negative' class having shorter texts and a smaller sample size, compared to the 'fair' class with larger texts and more samples, as well as 'positive' class with the most of everything. Several feature engineering strategies were attempted for baseline document-level sentiment analysis, but the most straightforward approach proved the most effective: to simply truncate all texts to the first 512 sub-words. Such a truncation is used for all sequence classification tasks.

## B Descriptive bias scores

| Model | N | F | M |
|---|---|---|---|
| NorBERT$_{3, x\text{-small}}$ | 2.19 | 31.01 | 4.15 |
| NorBERT$_{3, small}$ | 0.48 | 33.69 | 0.73 |
| NorBERT$_{3, base}$ | 1.22 | 33.21 | 5.00 |
| NorBERT$_{3, large}$ | 1.70 | 33.33 | 7.69 |
| mBERT | **3.41** | 6.47 | **31.99** |
| ScandiBERT | 0.97 | 16.23 | 26.73 |
| XLM-R$_{base}$ | 1.70 | 23.32 | 11.96 |
| XLM-R$_{large}$ | 2.07 | 18.07 | 26.49 |
| NorBERT$_{3, base}$, oversampled | 0.36 | 33.45 | 3.17 |
| NorBERT$_{3, base}$, NAK only | 2.56 | 28.81 | 18.43 |
| NorBERT$_{3, base}$, NCC only | 2.19 | 30.76 | 15.87 |
| NorBERT$_{3, base}$, mC4 only | 0.61 | 33.33 | 5.37 |
| NorBERT$_{3, base}$, NB only | 0.73 | 30.03 | 7.44 |
| NorBERT$_{3, base}$, Wiki only | 2.56 | 25.88 | 21.97 |
| NorT5$_{x\text{-small}}$ | 0.48 | 32.71 | 0.48 |
| NorT5$_{small}$ | 0.0 | **34.06** | 0.0 |
| NorT5$_{base}$ | 0.36 | 16.97 | 26.49 |
| NorT5$_{large}$ | 0.12 | **34.06** | 0.0 |

Table 8: Descriptive bias scores of gender-dominated and gender-neutral occupations. Where N stands for neutral, F for female, and M for male. Best score are typeset in bold, and worst scores are underlined.

## C   Hyperparameters

| Hyperparameter | NorBERT₃, x-small / small / base / large |
|---|---|
| Number of layers | 12 / 12 / 12 / 24 |
| Hidden size | 192 / 384 / 768 / 1 024 |
| FF intermediate size | 512 / 1 024 / 2 048 / 2 730 |
| Vocabulary size | 50 000 |
| Attention heads | 3 / 6 / 12 / 16 |
| Dropout | 0.1 |
| Attention dropout | 0.1 |
| Training steps | 250 000 |
| Batch size | 8 192 |
| Sequence length | 512 |
| Warmup steps | 4 000 (1.6% steps) |
| Initial learning rate | 0.01 |
| Final learning rate | 0.001 |
| Learning rate decay | cosine |
| Weight decay | 0.1 |
| Layer norm $\epsilon$ | 1e-7 |
| Optimizer | LAMB |
| LAMB $\epsilon$ | 1e-6 |
| LAMB $\beta_1$ | 0.9 |
| LAMB $\beta_2$ | 0.98 |
| Gradient clipping | 2.0 |

Table 9: Pre-training hyperparameters. The models differ only in their hidden size and number of layers, the learning rate schedule and other training settings are kept identical.

| Hyperparameter | Value |
|---|---|
| Dropout | 0.1 |
| Attention dropout | 0.1 |
| Label smoothing | 0.1 |
| Epochs | 10 |
| Max length | 512 |
| Batch size | 32 |
| Warmup steps | 250 |
| Initial learning rate | 0.001 |
| Final learning rate | 0.0001 |
| Learning rate decay | cosine |
| Weight decay | 0.1 |
| Optimizer | AdamW |
| Gradient clipping | 10.0 |

Table 10: Hyperparameters for fine-tuning language models on UD tasks.

| Hyperparameter | Value |
|---|---|
| Dropout | 0.1 |
| Attention dropout | 0.1 |
| Epochs | 10 |
| Max length | 512 |
| Batch size | 32 |
| Learning rate | 5e-5 |
| Learning rate decay | constant |
| Weight decay | 0.01 |
| Optimizer | AdamW |

Table 11: Hyperparameters for fine-tuning language models on NER and TSA.

| Hyperparameter | Value |
|---|---|
| Dropout | 0.1 |
| Attention dropout | 0.1 |
| Epochs | 10 |
| Max length | 512 |
| Batch size | 16 |
| Initial learning rate | 1e-5 |
| Learning rate decay | constant |
| Weight decay | 0.01 |
| Optimizer | AdamW |

Table 12: Hyperparameters for fine-tuning language models on document-level and sentence-level sentiment analysis.

| Hyperparameter | Value |
| --- | --- |
| Dropout | 0.1 |
| Attention dropout | 0.1 |
| Epochs | 10 |
| Max length | 512 |
| Batch size | 32 |
| Warmup portion | 6% |
| Initial learning rate | 1e-5 |
| Final learning rate | 1e-6 |
| Learning rate decay | cosine |
| Weight decay | 0.01 |
| Optimizer | AdamW |

Table 13: Hyperparameters for fine-tuning language models on NoCoLA.

| Hyperparameter | Value |
| --- | --- |
| Dropout | 0.0 |
| Attention dropout | 0.0 |
| Epochs | 10 |
| Max length | 512 |
| Batch size | 32 |
| Warmup portion | 6% |
| Initial learning rate | 2e-5 |
| Final learning rate | 2e-6 |
| Learning rate decay | cosine |
| Weight decay | 0.1 |
| Optimizer | AdamW |

Table 14: Hyperparameters for fine-tuning language models on Bokmål–Nynorsk machine translation.

| Hyperparameter | Value |
| --- | --- |
| Dropout | 0.1 |
| Attention dropout | 0.1 |
| Epochs | 3 |
| Batch size | 16 |
| Warmup steps | 100 |
| Max length | 384 |
| Document stride | 128 |
| Initial learning rate | 1e-4 |
| Final learning rate | 0.0 |
| Learning rate decay | linear |
| Weight decay | 0.01 |
| Optimizer | AdamW |

Table 15: Hyperparameters for fine-tuning language models on NorQuAD