# OpenReview forum: "NorBench -- A Benchmark for Norwegian Language Models"
_NoDaLiDa/2023/Conference — NoDaLiDa 2023_

### Official Review · Reviewer_ujwz · 2023-03-08
**NorBench or NorBERT_3 - what is the main contribution?**

**Rating:** 6
**Confidence:** 5

**Review:**

This paper presents a test suite for evaluating Norwegian language models called NorBench. The paper also presents NorBERT_3, which is a series of new Norwegian BERT-models. Roughly half of the paper is dedicated to NorBench, and half is dedicated to NorBERT. It is of course perfectly fine to have many different contributions in one paper, but in this case, the result is that none of the contributions are covered in sufficient detail. The effect is that the reader is left wondering what is intended to be the main contribution of the paper: is it NorBench or NoRBERT_3?

In my opinion, the most valuable contribution of this paper is the NorBench benchmark, which seems like a useful resource for evaluating and comparing Norwegian language models. However, I would expect much more discussion and analysis of the usefulness of the various tests in the benchmark. For example, most of the token-level tasks (in particular UPOS, UFeats and Lemma) show very small variance over the various models, which raises the question how useful such tests are. It would have been interesting to see results from a language model trained on a completely different language (e.g. Finnish) to ensure that the benchmark can actually discriminate between useful and not so useful models for Norwegian. One suggestion could be to add a row with the variance over all models for each test in the benchmark (Table 3), and to add some discussion about what this might show.

It would also have been interesting to see and compare the results from using some other benchmarks, such as ScanEval, to compare Norwegian models. You seem to be a bit critical of ScandEval, so one would expect that NorBench gives a better discrimination between useful and not so useful Norwegian models.

There are some interesting anomalies in the results that I think deserve more attention. For example, the relatively low scores for the NorBERT_3 models for the TSA test in Table 3. I think this is interesting, and would expect more discussion about this. The same goes for the surprisingly high score for NoRBERT_3,x-small for the Normative test in Table 4. I also wonder why you did not include the multilingual models in Table 4 and 5? I think this would be an interesting comparison. (And I think the exclusion of the multilingual models from these tables add to the impression that you are not so much evaluating the benchmark here as you are evaluating your new models.)

Overall, I think a paper on NorBench deserves to be published, but I would strongly suggest to refine the contribution of the paper to put more emphasis on NorBench rather than NorBERT_3.

**Paper Type:**

Long paper

---

### Official Review · Reviewer_V3Nv · 2023-03-10
**Review of ‘NorBench – A Benchmark for Norwegian Language Models’**

**Rating:** 8
**Confidence:** 4

**Review:**

This paper makes three substantial contributions:

1. It presents a suite of NLP tasks for evaluating Norwegian language models, dubbed NorBench.
2. It presents a new family of BERT-like Norwegian language models (NorBERT₃), trained according to best practices.
3. It evaluates the new models on the benchmark and evaluates various decisions taken for training, such as the choice of the data set.

## Strengths

1. The NorBench suite and the new version of the language models have significant value as NLP resources and for evaluation.
2. The findings from the evaluation, in particular the finding about the competitiveness of the Wikipedia-based models, should be interesting for all researchers developing language models for low-resource languages.
3. The paper is detailed and clearly written.

## Weaknesses

1. The authors do not motivate the specific task selection in their benchmark. NorBench has a clear bias towards general tasks such as morpho-syntactic analysis and classification, while other benchmarks (including the cited SuperGLUE) focus more on application-specific tasks such as inference or question answering. Why did the authors choose the tasks they chose? In what sense is the comparison facilitated by NorBench ‘fair’, as the authors claim in their description of their contributions. This reviewer would claim that NorBench is well-suited for the evaluation of encoder-based language models such as NorBERT, but perhaps less suited for the evaluation of other language models, including generative models.

2. I did not understand how the bias scores are calculated, or what exactly they measure. The authors write: ‘The normative score compares the gender-based aggregated probability distributions of templates with a normative distribution of genders in occupations. The idea here is that genders should be equally represented in each occupation with a gender distribution falling between 45% and 55% for each occupation.’ But how exactly are these scores ‘compared’? What is a ‘perfect score’? How can we conclude that ‘all models are much better at identifying female-dominated occupations’?

3. The authors do not discuss the extremely low result of their largest NorBERT₃ model on targeted sentiment analysis.


**Paper Type:**

Long paper

---

### Official Review · Reviewer_BPB4 · 2023-03-13
**This is a good paper, providing new language models and substantial new results on several Norwegian NLP tasks.**

**Rating:** 9
**Confidence:** 4

**Review:**

This paper presents a benchmark for Norwegian language models. The authors also present several new masked language models for Norwegian and provide a thorough comparison of various Norwegian NLP tasks on several previously existing Norwegian models and the new models trained as part of this paper.

The paper is well written and easy to read. The results presented offer new insights into how well factors such as the size of the LM and the data it has been trained on affects the performance of the downstream tasks. Although the content is not super original, the new models are definitely useful for others working on Norwegian NLP and the presented results can offer useful guidelines also for languages other than Norwegian.

Pros:
* Several new Norwegian language models with varying size and different training data sources are benchmarked on several NLP tasks, offering a thorough overview about how these models perform on these tasks.
* Depending on how the benchmark is implemented, it can be a useful resource for other people developing NLP tools for Norwegian.
* The paper is well written and easy to follow.

Cons:
* The benchmark seems to be limited to non-generative tasks only.
* The amount of resources spent on producing the results presented in the paper seems to be considerable.

Questions:
* Do you plan to limit the benchmark only to non-generative tasks or do you envision extending NorBench also to the generative domain?
* Can you give an indication of the resources spent on this work, both in terms of GPU hours and the total cost in €?

Other:
* Section 3.3 lists some training details (epochs, batch size, learning rate) and it remains unclear if this applies only to the targeted sentiment analysis task or to all benchmark tasks. If it applies only to the targeted sentiment analysis task then what are the respective details for other tasks? If this information applies to all tasks then this material should be probably presented somewhere else.
* In line 132 the word “be” seems to be missing.

**Paper Type:**

Long paper

---

### Decision · Program_Chairs · 2023-03-17

Accept